# Accuracy of HemoCue301 portable hemoglobin analyzer for anemia screening in capillary blood from women of reproductive age in a deprived region of Northern Peru: An on-field study

Dulce E. Alarcón-Yaquetto[1]*, Lenin Rueda-Torres[1], Nataly Bailon[1], Percy Vílchez Barreto[2], Germán Málaga[1]

1 Unidad de Conocimiento y Evidencia (CONEVID), Universidad Peruana Cayetano Heredia, Lima, Perú,
2 Centro de Salud Global, Tumbes, Peru

☯ These authors contributed equally to this work.
* dulce.alarcon@upch.pe

**Data Availability Statement:** Data is available at https://osf.io/4hrpx/.

## Abstract

### Objective

We aim to assess the accuracy and effectiveness of the HemoCue 301, a point-of-care (POC) device for measuring hemoglobin levels, and detecting anemia among individuals living in Tumbes, a rural, underserved area in Northern Peru.

### Methods

Baseline analysis of a clinical trial aimed at assessing the effect of multi-fortified bread (NCT05103709). Adult women with capillary blood HemoCue 301 readings below 12 g/dL were recruited in coastal city of Tumbes, Peru. A total of 306 women took part of the study, venous blood samples were taken and analyzed with an automated hematology analyzer. Serum samples were used to measure ferritin, serum iron and C reactive protein.

### Results

Capillary blood measured by the Hemocue 301 has a bias of 0.36 ± 0.93 g/dL respect to the automated Hb. More than 50% of women with normal ferritin values were classified as anemics according to the HemoCue 301. Automated Hb cut-off of 10.8 g/dL [AUC 0.82 (0.77–0.88)] had a specificity of 0.817 and a sensitivity 0.711 while with the HemoCue 301 cut-off of 11.1 g/dL [AUC 0.71 (0.62–0.79)] had a specificity of 0.697 and a sensitivity 0.688. The performance of the automated Hb cut-off was significantly better than the HemoCue (p<0.001).

**Funding:** The study was funded by CONEVID, Universidad Peruana Cayetano Heredia. The funders had no role in study design, data collection and analysis, decision to publish, or preparation of the manuscript.

**Competing interests:** The authors have declared that no competing interests exist.

## Conclusion

Caution must be taken when using POC devices, especially with values around the threshold. Cut-off values found in our study could be used as surrogate means when no confirmatory tests are available. Clinical outcomes should be prioritized when diagnosing iron deficiency anemia in women of reproductive age to ensure proper diagnosis.

## Introduction

Despite major breakthroughs in health technology, iron deficiency anemia (IDA) remains a challenge worldwide. Its long-standing association with deprivation and impoverishment has made IDA's eradication a universal goal to be achieved in the 2030 agenda [1]. A major hurdle to reach this objective is the inequity in its diagnosis with people from isolated communities being the more affected [2].

In Peru, anemia is considered a severe public health problem as shown by national statistics [3]. In some provinces, the rates of anemia greatly surpass the 40% cut-off value which is considered a critical threshold for public health concern, indicating a severe public health issue [4].

Still, research has questioned these figures due to several issues in the diagnosis. For instance, the sole measurement of hemoglobin (Hb) is not enough to diagnose IDA. It is widely known that Hb is only a proxy for IDA. Rationale behind its use is that ~70% of active bodily iron is contained in the protein [5]. However, it has even been showed to be poorly correlated to iron deficiency *per se* [6]. Despite this fact, the main strategy of health policies aimed to tackle anemia in Peru are based on iron supplementation [7]. IDA diagnosis needs the estimation of iron parameters and should not only rely on Hb measurement. This over simplification might be counterproductive [8].

The need for further confirmatory tests that properly give an account of iron homeostasis in the organism is warranted. These might be readily available in high-income countries and even in main cities from low-and-middle income countries (LMIC) but hardly ever offered to rural and remote populations.

Furthermore, automated hematology analyzers—the method suggested by the WHO [9]—cannot be properly used in remote communities due to resource constraints. Handheld devices have been routinely used to screen for anemia in LMIC countries such as the HemoCue system. In fact, this device is used by the Peruvian government as the main method for anemia surveillance, and public policy making [10].

The HemoCue 301 system is the newer version of the point of care device. The microcuvettes are not that sensitive to extreme weathers as the previous versions (201 and 201+) and the cost per sample is cheaper [11, 12]. While the manufacturer states the device's discrepancy falls within 7% acceptable error compared to CLIA'88 regulations [13], Morris et al. (2007) found that 4% of samples were above the 7% limit and 1% above 10% error in blood-donor samples from the UK [12]; Rappaport et al. (2017) found a bias of 2 g/L compared to the 201 version in Cambodian women of reproductive age (WRA) [14] and Yadav et al. (2020) found a -0.25 (0.85) g/dL bias compared to automated analyzer in pregnant women from India [15].

Here, we compare the performance of HemoCue 301 to the reference method, automated hematology analyzer, while also assessing iron status in Tumbes, Peru. The main objective of this study is to assess the suitability and accuracy of this device as a method for anemia diagnosis in women of reproductive age in a coastal city with resource constraints.

## Methods

### Design and setting

Baseline analysis of a clinical trial aimed at assessing the effect of the consumption of multi-fortified bread in WRA with iron deficiency anemia. The study was carried out in Tumbes, a coastal city of Northern Peru with altitudes ranging from 6 to 124 meters above sea level. According to official statistics, Tumbes has between 9.2 to 12% of poverty and mean *per capita* earnings of 558 soles (~150 USD), 13.61% less than the national average [16].

### Inclusion criteria and point of care Hb measurement

The study recruited apparently healthy, non-pregnant WRA, living permanently in the city. Inclusion criteria was adult WRA, living permanently in the city with capillary blood Hemocue301 readings below 12 g/dL. Women with amenorrhea non explained by pregnancy and those who were heavy smokers were excluded from the study. Due to ethical concerns raised by the IRB, all women with HemoCue values below 8 g/dL were also excluded and referred to primary care centers. Recruitment took place between August 5th and September 17th 2021.

HemoCue measurement was done by trained nurses following the manufacturers recommendations. Briefly, 301 cuvettes were filled with a drop of blood from the middle finger fingertip. The first 2 drops of blood were discarded. The microcuvette was put in the cuvette holder and a measurement was done immediately [11]. A trained phlebotomist took venous samples of each woman with the BD vacutainer system (one tube with EDTA and other without anticoagulant). Both HemoCue measurement and venous samples were taken the same day and time and at each participant's house after informed consent was signed.

### Laboratory analysis

Whole blood samples in EDTA tubes were transported to the Tumbes Regional Direction of Health in a temperature monitored cooler that remained at laboratory temperature (18˚- 24˚ C). A complete blood count (CBC) was performed in venous blood samples within two hours of obtaining the sample. The CBC analyses followed the ISLH´s guidelines [17]. Briefly, slide review was performed to assess morphological changes of the red blood cell. No features suggestive of thalassemia or other type of hemoglobinopathies were found. An automated analyzer Prokan PE-6100 PLUS was used to measure hematological indices within the 4 hours after the sample was taken. A second tube with silica clot activator was transported in coolers at 4˚ - 8˚C to the Tumbes Regional Direction where they were centrifuged, remaining serum was stored at -20˚C and shipped to Lima where C-reactive protein (CRP), serum iron and ferritin were measured by spectrophotometric and turbidimetric assays (Wiener Lab CM250, Wiener Lab Switzerland) on an ISO 15189 compliant laboratory.

### Statistical analysis

Anemia was defined using automated Hb values of 12 g/dL. Iron deficiency was assessed using ferritin values below 15 ug/dL [18] and IDA was defined when both conditions were satisfied (automated Hb values below 12 g/dL and ferritin below 15 ug/dL). Microcytosis was defined with mean corpuscular values (MCV) < 80 fl., hypochromia by mean corpuscular hemoglobin (MCH) values below 27. Descriptive statistics are provided as mean standard deviation or median and interquartile range for quantitative variables according to their distribution. Frequency and percentages are used for qualitative variables. The population was further divided into tertiles based both on automated Hb values and HemoCue values. Serum iron, ferritin

and CRP were compared between tertiles, as well as other red blood markers and analysis of variance was used to assess differences.

Bland Altman plots are presented between automated Hb (aHb) and Hb values given by the HemoCue301. aHb is used as gold standard as it is the measure suggested by the WHO [9]. The plots were constructed using the B.A function [19] in RStudio. The normality of the differences was assessed through a Shapiro Wilk test.

Low ferritin values were defined as serum ferritin values below 15 ug/L [18]. The accuracy of the HemoCue Hb and automated Hb to identify low ferritin values was assessed using receiver-operating characteristic (ROC) curves. For this, a general linear model adjusted by age was plotted. ROC curves were constructed using the pROC package [20] using low ferritin (<15 ug/L) as gold standard. Ferritin shows the major diagnostic accuracy for iron deficiency in relation to bone marrow aspirate [21]. A sensitivity analysis was performed excluding all those participants with CRP values above 5 mg/L. Statistical test for ROC curves was the DeLong test. Optimal cut-off values were calculated using the Youden index. Sensitivity, specificity, positive and negative predictive values and their respective confidence intervals are provided for each cut-off point. Statistical significance was denoted by p values below 0.05. Analyses were performed with RStudio©1.1.45327(RStudio, Boston, MA).

## Results

A total of 306 women participated in the study with an average age of 25.94±5.4 years. While according to our inclusion criteria, upon enrolment, all women had HemoCue readings below 12 g/dL, the median automated Hb values were 11.7 (10.6–12.3) g/dL. 123 women (40.2%) had automated Hb values below 12 g/dL, however, out of this group only 43 (23.6%) had also ferritin values below the threshold. While not the only reason of microcytic and hypochromic anemia, IDA is characterized by microcytosis and hypochromosis. Out of the sample, 53.95% had microcytosis while hypochromia based on MCH was found in 37.5%. Both conditions were found concomitantly in 34.21% of the sample. Table 1 features sociodemographic variables according to automated Hb status.

We further classified participants based on tertiles of both capillary blood HemoCue 301 values and venous blood automated Hb values. The lowest Hemocue301 tertile had HemoCue values ranging from 8–11 g/dL (n = 111). In the second group HemoCue values ranged from 11.1 to 11.5 g/dL (n = 102), while the final group had values from 11.6 to 11.9 g/dL (n = 93).

In Fig 1, we show differences between red blood cell count indices of relevance for IDA diagnosis. The main difference resides between parameters from the first and second tertiles and with the first and third tertiles. No significant difference is seen between the second or the third tertile of any of the markers seen suggesting the main hurdle for the HemoCue 301 is distinguishing people with values close to the cut-off of 12 g/dL. Automated Hb values in the

**Table 1. Baseline characteristics of sample by automated Hb status.**

| Variable | Total (n = 306) | aHb> = 12 (n = 123) | aHb<12 (n = 183) | P value |
|---|---|---|---|---|
| Age (years) | 25.5 (21–31) | 26 (22–31) | 25 (20–31) | 0.302 |
| Height (meters) | 1.55±0.06 | 1.54±0.05 | 11.55±0.06 | 0.053 |
| Weight (kilograms) | 62.6 (54.55–76.15) | 64.4 (54.4–75.1) | 62.1 (54.7–76.5) | 0.838 |
| BMI (m/kg$^2$) | 26.66 (22.81–30.86) | 27.58 (23.13–31.56) | 26.37 (22.59–30.57) | 0.333 |
| HemoCue Hb (g/dL) | 11.3 (10.8–11.6) | 11.5 (11.2–11.7) | 11.1 (10.4–11.5) | <0.001 |

Hb: Hemoglobin, aHb: Hb measured by automated analyzer. Values are mean ± SD or median and IQR. P values are from 2 tailed T tests or Wilcoxon Rank sum test.

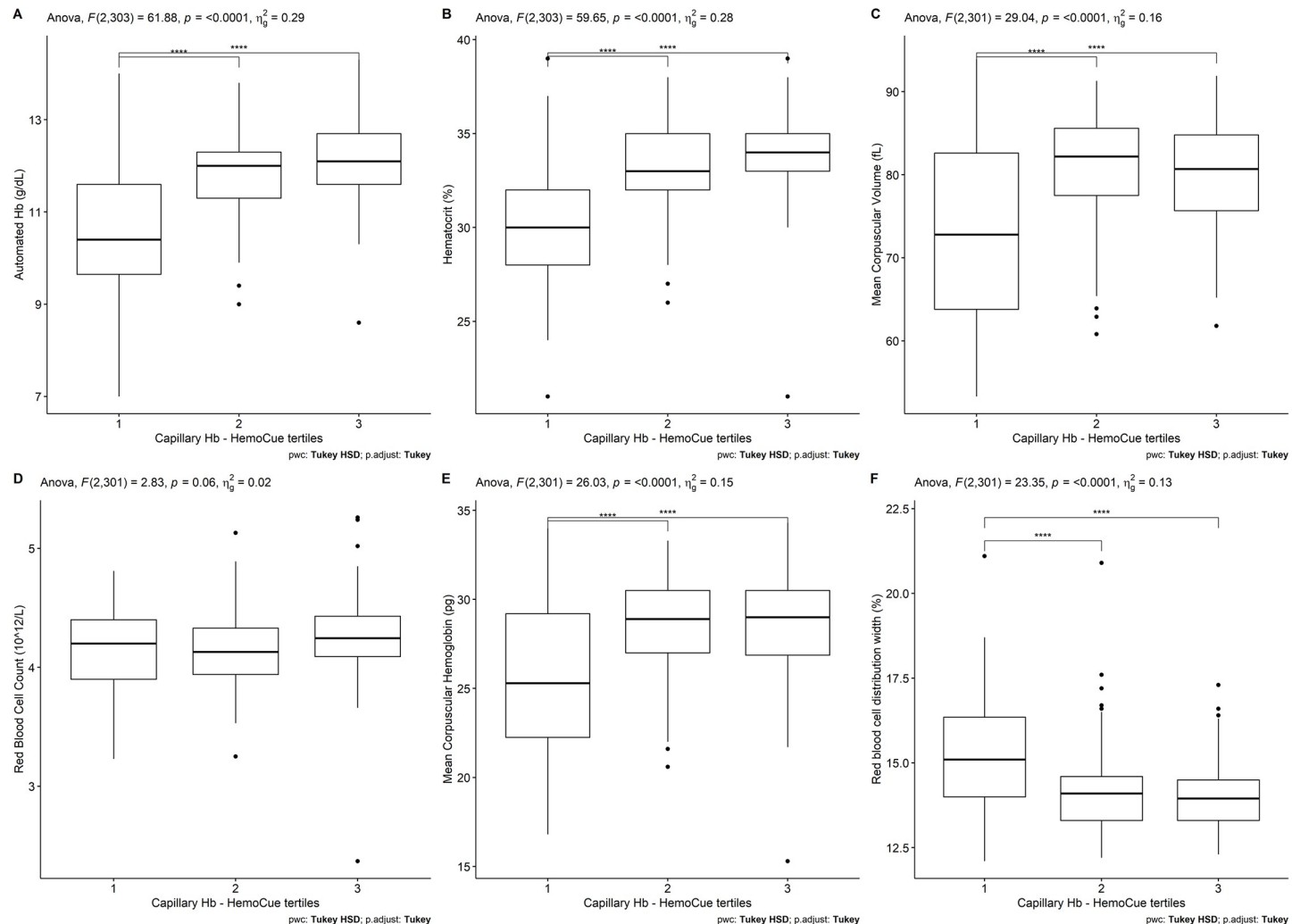

**Fig 1. Red blood cell indices according to HemoCue hemoglobin tertiles.** A) Automated Hb, B) Hematocrit, C) Mean Corpuscular Volume, D) Red Blood Cell Count, E) Mean corspuscular hemoglobin, F) Red blood cell distribution width ****p<0.001.

highest tertile were 12.15±0.89 g/dL (Fig 1A). In the case of red blood cell count, no difference is seen between any of the tertiles (Fig 1D).

Fig 2 shows the same red blood cell indices but according to automated Hb tertiles. Based on this measure, 104 participants were categorized on the lower tertile with aHb values ranging from 7–11.1 g/dL, 110 were categorized on the second tertile (11.2–12.1 g/dL) and the highest tertile was comprised by 92 participants (12.3–14.3 g/dL). Differences between the tertiles are more pronounced than when dividing the population in automated Hb tertiles. For instance, hematocrit, MCV and MCH had all significant difference between each group. RDW values were only different between the two lowest tertiles and between the lowest and the highest tertile. No difference was seen in RBC values (Fig 2).

In Table 2, we present the prevalence of anemia according to each method and the severity. Given our inclusion criteria, we did not recruit women with HemoCue values above 12 g/dl or below 8 g/dL. Therefore, these rates must be taken with caution.

Iron status biomarkers were also assessed between tertiles as shown in Table 3. Ferritin is significantly different in all tertiles when using automated Hb to classify, but when HemoCue

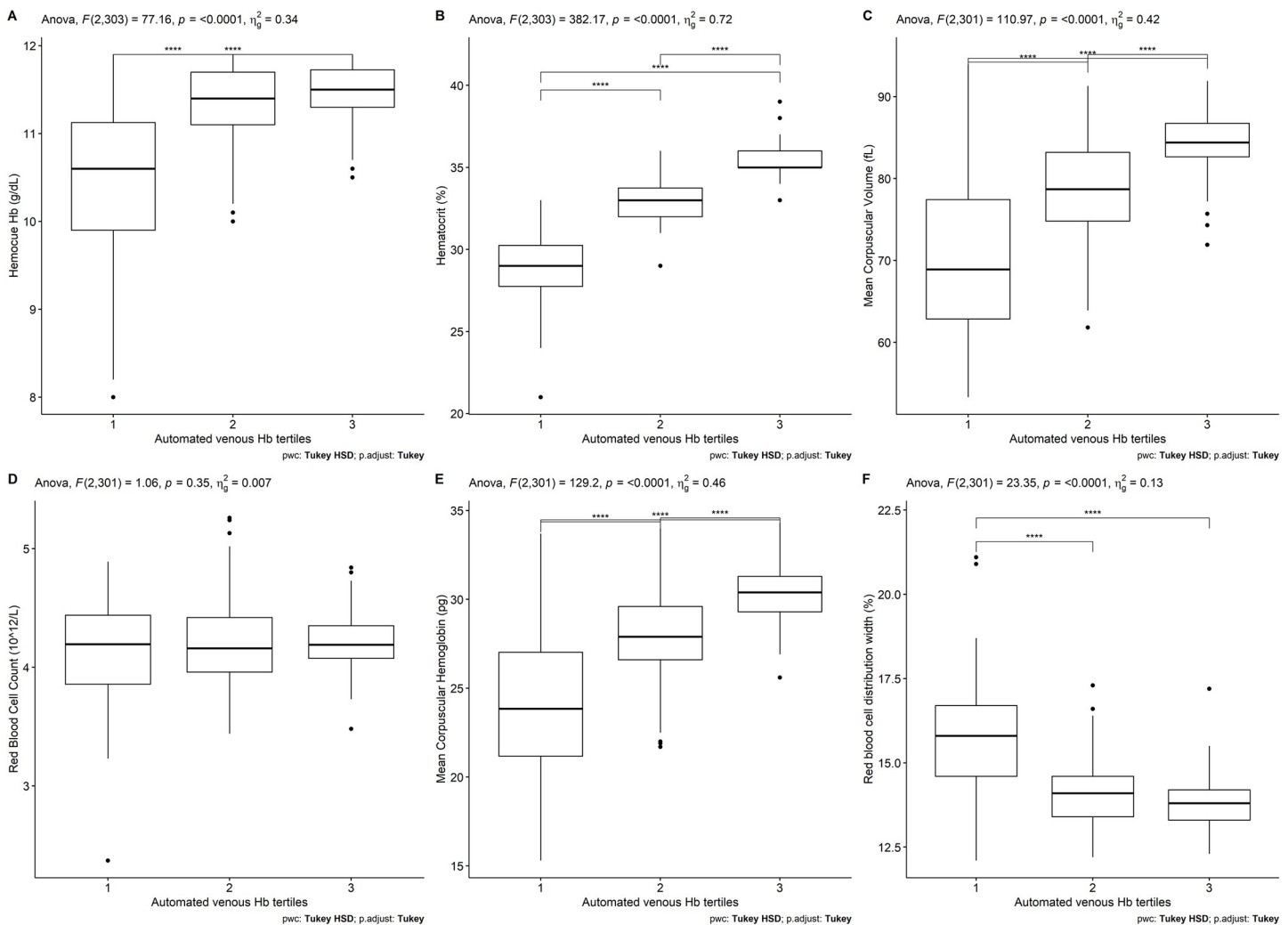

**Fig 2. Boxplots of red cell blood indices according to automated hemoglobin tertiles.** A)HemoCue Hb, B) Hematocrit, C) Mean Corpuscular Volume, D) Red Blood Cell Count, E) Mean corspuscular hemoglobin, F) Red blood cell distribution width. **** p<0.001.

Hb values are used, there is no difference between the 2nd and 3rd tertiles. The same pattern is observed with serum iron. CRP levels are not significantly different in any classification.

In the whole sample, 182 women had venous blood—automated Hb values below 12 g/dL. However, out of this number, only 43 (23.63%) had also ferritin values under 15 ug/L. When excluding those with possible inflammation (n = 215, CRP> 5 mg/L), 129 women had anemia of which only 35 (27.13%) had also low ferritin values. On this group, 94 women (52.5%) with normal ferritin values are regarded as anemics.The degree of agreement between automated

**Table 2. Prevalence and severity of anemia according to different methods.**

|  | Capillary blood HemoCue 301 | Venous blood Automated Analyzer |
| --- | --- | --- |
| No anemia (> 12 g/dL) | – | 41.47% |
| Mild anemia (11.-11.9 g/dL) | 67.01% | 30.77% |
| Moderate anemia (8–10.9 g/dL) | 32.9% | 26.76% |
| Severe anemia (< 8 g/dL) | – | 1% |

**Table 3. Iron status markers according to HemoCue and automated hemoglobin tertiles.**

| | HemoCue tertiles | | |
|---|---|---|---|
| **Marker** | I tertile (Hb 8–11.0) | II tertile (Hb 11.1–11.5) | III tertile (Hb 11.6–11.9) |
| **Ferritin (ug/L)[a, b, c]** | 20.53 (11.4–33.09) | 32.57 (20.71–49.4) | 29.51 (20.22–46.26) |
| **Serum iron (ug/dL) [c, d, e]** | 49.55±33.95 | 62.88±32.59 | 72.08±31.73 |
| **CRP (mg/L)[f]** | 3.61±5.47 | 4.41±5.44 | 4.21±3.97 |
| | Automated Hb tertiles | | |
| **Marker** | I tertile (Hb 7–11.1) | II tertile (Hb 11.2–12.1) | III tertile (Hb 12.3–14.3) |
| **Ferritin (ug/L)[g]** | 18.73 (11.05–29.5) | 25.62 (19.33–37.5) | 43.3 (31.06–57.51) |
| **Serum iron (ug/dL)[g]** | 38.63±27.23 | 62.21±27.69 | 84.21±31.60 |
| **CRP (mg/L)[f]** | 4.15±6.64 | 4.35±4.49 | 3.61±3.32 |

Values are mean ±SD. P values are from Tukey post hoc analysis [a] p = 0.002 I vs II tertile.

[b] p = 0.008 I vs III

[c] not significant II vs III tertile

[d] p = 0.01 I vs II tertile

[e] p<0.0001 I vs III tertile

[f] not significant between any groups

[g] p<0.0001 between all groups. CRP: C-reactive protein.

Hb values from venous blood samples and capillary blood HemoCue values is shown in a Bland Altman Plot (Fig 3). The bias between measurements is 0.36 ± 0.93 g/dL with the automated Hb yielding higher values. The 95% CI for the measurements was -1.47–2.19. The differences were normally distributed (Shapiro Wilk Z = 0.296, p = 0.3834).

The ROC curve in Fig 4 shows the performance of both methods compared to ferritin measurements. We performed the analysis in the whole sample (Fig 4A) and excluding 90 subjects whose CRP values exceeded 5 mg/L. Automated Hb values are the ones showing better performance in the whole sample [AUC 0.82 (0.77–0.88) vs 0.71 (0.62–0.79), p<0.001] and in those with no suspected inflammation [AUC 0.83 (0.77–0.89) vs 0.70 (0.77–0.89), p<z0.001]. We calculated the best cut-off value for each measurement using the Youden Index. The cut-offs for each diagnostic test are provided in Table 4 alongside their specificity, sensitivity and predictive values. Using the Youden index, the optimal aHb cut-off is 10.8 which has a specificity of 0.711 and a sensitivity of 0.819, while the Hemocue reaches a specificity of 0.697 and a sensitivity of 0.688 with a cut-off of 11.1 g/dL. Positive and negative predictive values show that overall automated Hb with a cut-off of 10.8 g/dL accurately identifies 94% of patients that have low ferritin values. The cut-off values provided could be useful where no other means or confirmatory tests are available.

## Discussion

A right-based approach to health encompasses various characteristics, including accessibility, availability, quality, and equity. Ensuring that goods and services are of high quality and distributed equitably is essential, with no disparities based on geographic location or socio-economic status [22]. In this study we sought to determine whether anemia screening using the HemoCue 301 portable device was suitable in a high poverty area on the northern coast of Peru.

As we have seen, HemoCue 301 Hb values are significantly lower than those yielded by the automated method. The difference is of clinical relevance, 0.3 g/dL might determine if a woman is classified as anemic or not and misclassifying the severity of the condition.

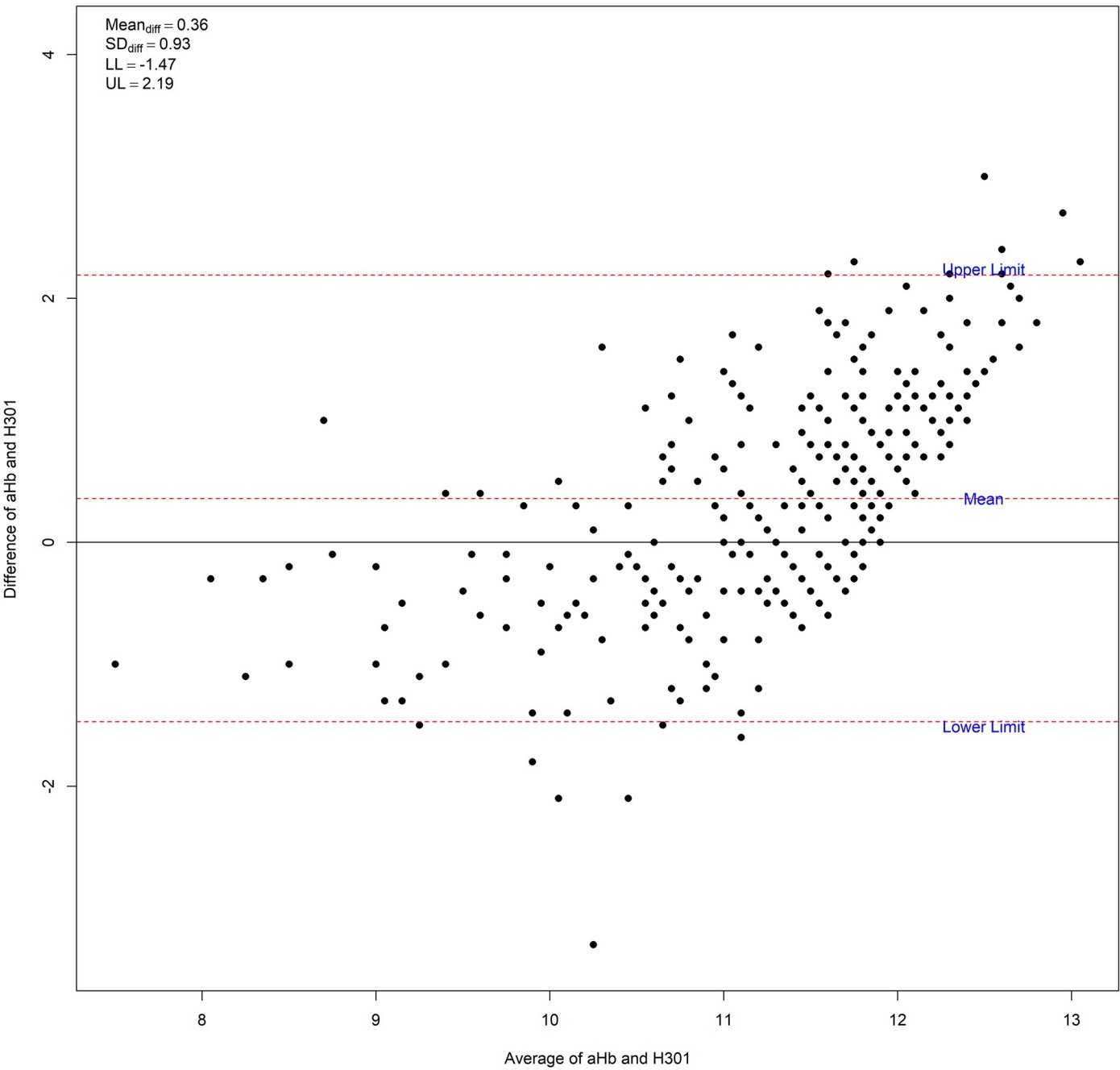

**Fig 3. Bland Altman limits of agreement plot between automated Hb and HemoCue 301.** UL: Upper limit, LL: lower limit.

HemoCue 301 is a screening device and it should be taken as such. Screening and diagnostic testing are two different procedures linked between each other. Screening is aimed at identifying those in high risk, and then, a diagnostic testing aims to provide a definite answer to whether or not the person has a given condition to start a treatment [19]. The analysis of other markers of IDA as CBC indices, ferritin, and serum iron allow us to see that using HemoCue 301 as a diagnostic device is not correct as it does not accurately classify subject with Hb values

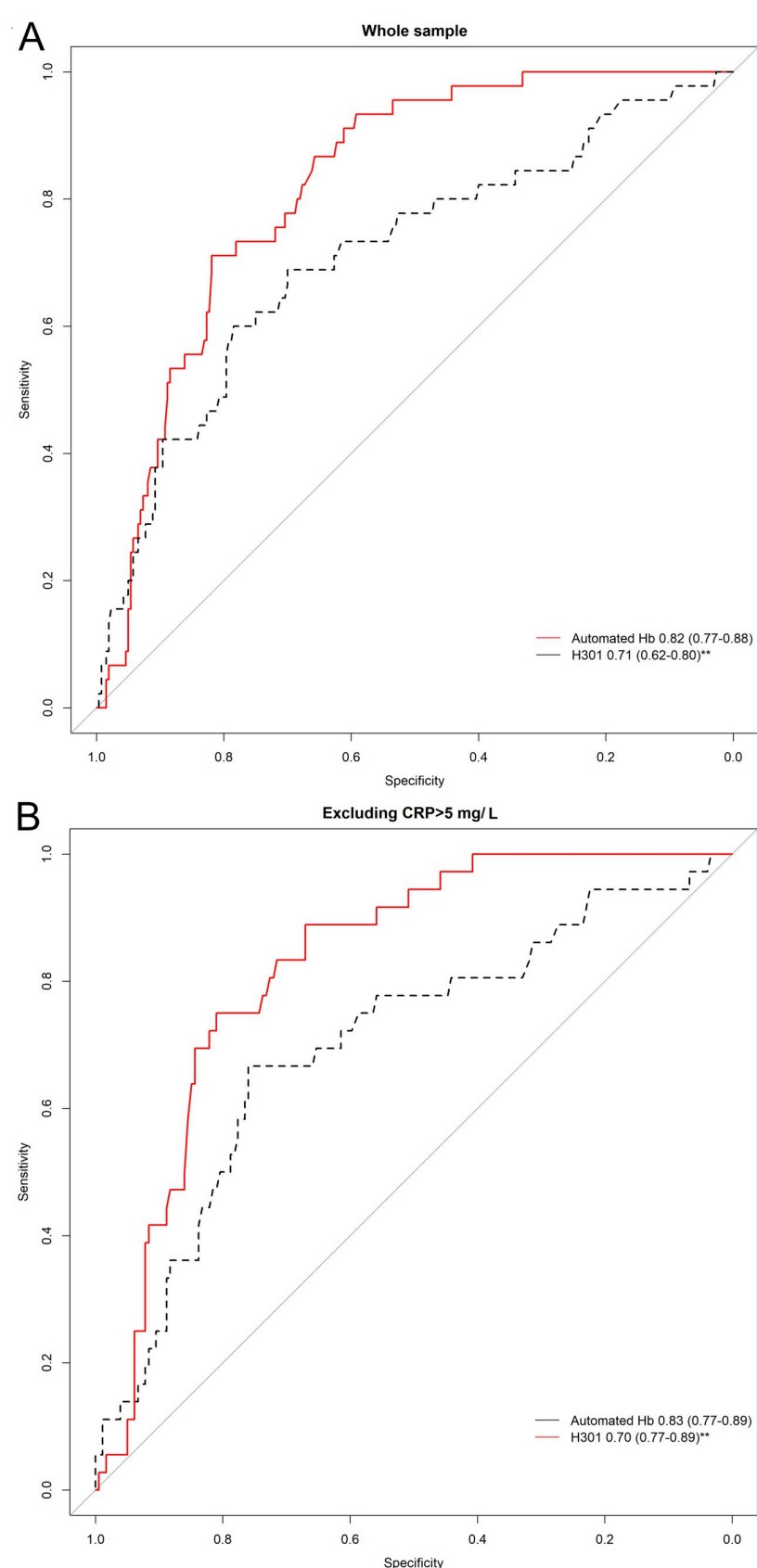

**Fig 4. Are under the ROC curve for hemoglobin measured with the Hemocue301 and the automated analyser (aHb).** The gold standard used was ferritin values below 15 ug/ml to define IDA.

close to the threshold (12 g/dL). Women with ferritin values above 100 ug/L which generally excludes iron deficiency anemia [23] were regarded as anemics when using the POC device.

The cut-offs to define anemia have been a matter of debate since they were first proposed. Mainly because the basis of their original determination were statistics rather than health outcomes [24]. In a recent multi-country analysis, the pooled 5[th] percentile of apparently healthy WRA was 10.81 (10.35–11.17) [25]. Furthermore, mild anemia in WRA (11–11.9 g/dL) was found not to be associated with small for gestational age in pregnant women in a systematic review [26]. The U-shaped association of hemoglobin values and adverse pregnancy outcomes has been established in several studies [27, 28] and a revision based on adverse outcomes has been suggested.

In our study, we find that 10.8 g/dL of venous blood samples analyzed with an automated hematological analyzer is the optimal cut-off value in terms of specificity, sensitivity and positive predictive value. Negative predictive value is around 40% which is acceptable for a screening test. Capillary blood Hemocue cut-off of 11.1 g/dL is a better predictor than the current cut-off of 12 g/dL and might be suitable in cases where measuring iron biomarkers is not possible.

Our results are in line with other studies. In Laotian children, Hemocue301 yielded lower Hb values than automated hematology analyzers [29]. The sensitivity found in this study for the HemoCue 301 is a little bit higher than that found in women in South Africa (0.72) [30] and Cameroon (0.62) [31]. As the authors of those papers state, it is a good performance for a screening device. Nevertheless, its use in Peru is not for the purpose of screening. These devices are used to quantify the total prevalence of anemia in the country; thus, inform public policies. Due to the high prevalence of anemia, preventive iron supplementation strategies are available in Peru [32, 33].

This approach would not signify further trouble if it were innocuous. Nevertheless, mounting evidence suggests it is not. Even mild iron overload is associated with an array of adverse outcomes including metabolic syndrome [34]. Furthermore, in deprived places where infections are endemic and sanitation is still an unmet need, iron supplementation might be counterproductive [35].

The benefits of the HemoCue cannot be denied. We support Sanchis-Gomar recommendations about when it is more needed. These circumstances include critical areas where a fast therapeutic decision is needed, or with patients in life-threatening conditions due to the low amount of blood needed as well as in natural disasters [36]. Our results suggest its use in population-based surveys is not appropriate.

Another argument used for the widespread deploy of HemoCue is the relative low-cost. Nevertheless, as far as we know, no proper health economic analysis has been performed to substantiate this claim. A CBC allows to screen for microcytosis, hypochromia and anisocytosis, which besides providing a better understanding as if the patient has IDA or not, is helpful

**Table 4. Cut-off values for automated analyzer and Hemocue 301.**

| | Cut-off value (g/dL) | Sensitivity (95% CI) | Specificity (95% CI) | Positive predictive value (95% CI) | Negative Predictive Value (95% CI) |
|---|---|---|---|---|---|
| Venous Blood–Automated analyzer | 10.8 | 0.82 (0.77–0.86) | 0.71 (0.56–0.84) | 0.94 (0.89–0.96) | 0.41 (0.33–0.59) |
| Capillary blood Hemocue 301 | 11.1 | 0.69 (0.63–0.74) | 0.67 (0.51–0.8) | 0.92 (0.86–0.94) | 0.27 (0.22–0.42) |

to identify other possible causes of anemia such as inflammation. Knowing the underlying cause of the low Hb count is essential to provide the best treatment possible. According to global estimates, only half of the cases of anemia worldwide are due to IDA, while around a 40% are due to inflammation [37, 38]. The fact that accessibility to an automated analyzer is still a challenge in cities from an upper middle-income country such as Peru is a clear example of the inequities in health, which -we consider- is the main reason Peru has stagnated in its quest to diminish anemia prevalence.

This study has several limitations. The original sample size for the clinical trial was 351 participants. However, the prevalence of WRA with Hemocue readings below 12 g/dL was significantly below our expectations. The recruitment was stopped when over 300 women were recruited. However, the sample size gathered is sufficiently powered to detect statistically significant differences in iron status biomarkers. Also, due to nature of the condition and it being considered a major global health problem in Peru, we could not extract venous blood samples from women with Hemocue values below 8 g/dL. The prevalence of anemia and its severity must be interpreted in line of this recruitment considerations.

Furthermore, we compared capillary blood to venous blood which are not equivalent in terms of hemoglobin determination. Although some research has found the difference not to be clinically significative [39], it is important to see our results in light of the difference in sampling sites. Finally, even when a peripheral blood smear was performed on samples to identify features suggestive of hemoglobinopathies, this method cannot be used to definitively rule out thalassaemia or other haemoglobinopathies.

Our study shows that 52.2% of women with normal ferritin values are classified as anemics using HemoCue 301 in a coastal city in Peru. Our results urge for caution when using HemoCue 301 device to diagnose and treat IDA, mainly when cases of mild anemia are found. HemoCue 301 should only be used for screening. We call for the use of better methods to diagnose iron deficiency anemia in Peru, especially in areas where settings inherently represent a limitation to the correct functioning of these devices, such as extreme temperatures.

## Acknowledgments

The authors would like to thank field workers at the *Centro de Salud Global*, that oversaw data collection in Tumbes. We also thank women who took part of the study.

## Author Contributions

**Conceptualization:** Dulce E. Alarcón-Yaquetto, Lenin Rueda-Torres, Germán Málaga.

**Formal analysis:** Dulce E. Alarcón-Yaquetto, Lenin Rueda-Torres.

**Funding acquisition:** Percy Vílchez Barreto.

**Investigation:** Dulce E. Alarcón-Yaquetto, Lenin Rueda-Torres, Nataly Bailon.

**Methodology:** Dulce E. Alarcón-Yaquetto, Lenin Rueda-Torres.

**Resources:** Percy Vílchez Barreto.

**Supervision:** Germán Málaga.

**Writing – original draft:** Dulce E. Alarcón-Yaquetto.

**Writing – review & editing:** Lenin Rueda-Torres, Nataly Bailon, Percy Vílchez Barreto, Germán Málaga.

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
