## [Decision Letter · Decision Letter 0]

14 Aug 2023

PONE-D-23-21513Accuracy of the Portable Hemoglobin Analyzer HemoCue301 in Women of Reproductive Age in a Deprived Region of Northern Peru: An On-Field StudyPLOS ONE

Dear Dr. Alarcón-Yaquetto,

Thank you for submitting your manuscript to PLOS ONE. After careful consideration, we feel that it has merit but does not fully meet PLOS ONE’s publication criteria as it currently stands. Therefore, we invite you to submit a revised version of the manuscript that addresses the points raised during the review process.

We look forward to receiving your revised manuscript.

Kind regards,

Benedikt Ley, PhD

Academic Editor

PLOS ONE

Journal Requirements:

Reviewers' comments:

Reviewer's Responses to Questions

**Comments to the Author**

1. Is the manuscript technically sound, and do the data support the conclusions?

Reviewer #1: Partly

Reviewer #2: Yes

2. Has the statistical analysis been performed appropriately and rigorously? 

Reviewer #1: No

Reviewer #2: Yes

3. Have the authors made all data underlying the findings in their manuscript fully available?

Reviewer #1: Yes

Reviewer #2: Yes

4. Is the manuscript presented in an intelligible fashion and written in standard English?

Reviewer #1: Yes

Reviewer #2: Yes

5. Review Comments to the Author

Reviewer #1: REVIEWER REPORT FOR MANUSCRIPT PONE-D-23-21513

General comment

Alarcón-Yaquetto et al have attempted to add data to the venous vrs capillary haemoglobin comparison study viz-a-viz different devices/instrument conundrum. However, the authors have not clearly defined basic concepts well to enable the reader to properly situate the findings. For example, the title is a bit misleading considering the study design employed. In line 86 – 88 (page 5), the authors write that “Briefly, 301 cuvettes were filled with a drop of blood from the middle finger fingertip. The first 2 drops of blood were discarded. The microcuvette was put in the cuvette holder and a measurement was done immediately”. In line 90 – 91 (page 5), the authors write that “Upon recruitment, a trained phlebotomist took venous samples of each woman with the BD vacutainer system”. The laboratory samples were venous blood samples. Therefore, the study compared venous haemoglobin (from automated analyser) to capillary haemoglobin (Haemocue 301). it is a known fact that venous haemoglobin and capillary haemoglobin are not equivalent (https://www.mdpi.com/2075-4418/12/12/3191; 10.1371/journal.pone.0278350). The title of the study is thus misleading and should be modified to reflect exactly what was done. Again, the distinction between iron deficiency anaemia and anaemia has not been made; the authors give the impression that the two terms are synonymous. However, that is not a scientific fact. This has confounded the write up and the data analyses as well as its interpretation. The manuscript should thus be revised holistically to address these.

Abstract

1. The abstract has been written on the basis of the assumptions that were used to analyse the data. As suggested above and below, the abstract need a re-work after taking the suggested revisions into consideration.

2. In the objective statement, indicate that the sampling site was different for the Haemocue and the automated haematology analyser.

Introduction

1. Line 40 – 41, authors seem to portray IDA as being equivalent to anaemia by the statement that “Peru is one of the countries where IDA is a severe public health problem as shown by national statistics.”

2. Line 56 – 57, authors write “Handheld devices have been routinely used to screen for IDA in LMIC countries such as the HemoCue system” to imply that Hemocue is a device for undertaking iron studies. This is misleading since the device estimates haemoglobin levels which is different from ferritin, iron, TIBC etc.

3. Authors again write that “The main objective of this study is to assess the suitability and accuracy of this device as a method for IDA diagnosis in women of reproductive age in a coastal city with resource constraints” Hemocue is not meant for iron studies.

Exclusion criteria

1. When authors write “Exclusion criterion was amenorrhea non explained by pregnancy and heavy smoking” does this mean that both conditions should be present before exclusion?

Statistical analyses

1. Authors defined Microcytosis using the CBC reported mean corpuscular values (MCV) < 80 fl.,

2. and hypochromia by mean corpuscular hemoglobin (MCH) values below 27. There are many causes of microcytic-hypochromic red blood cell picture. With the emphasis the authors are placing on iron deficiency anaemia, the authors should have at least undertaken peripheral blood morphology evaluation to assess whether there were unique red cell presentations such as pencil-shaped cells. This was not done.

3. Authors should clearly differentiate iron deficiency (ferritin less than 15 ng/dl) from iron deficiency anaemia (ferritin less than 15 ng/dl + Hb below threshold). This distinction has not been made. Since the authors make the erroneous assumption that IDA is equivalent to anaemia, it is not clear what the ROC analyses sought to predict.

4. Moreover, authors write that “Due to being an acute phase protein, ferritin is sensitive to inflammation. Values above 120 ug/L were excluded from the analysis.” The CRP cut-off of (5 mg/l) should be used to screen all the data but not only participants with ferritin levels >120 mcg/L. It is even possible that some with ferritin levels >120 mcg/L had haemochromatosis instead of inflammation-induced increased ferritin levels. Therefore, the assumption of excluding data only based on the ferritin levels >120 mcg/L is erroneous and should be corrected.

Results

1. The authors write that “123 women (40.2%) had ferritin values 125 above 15 g/dL using the automated analyzer”. What does this represent? ID or anaemia? The two are not equivalent and must be acknowledged as such. The haematotology anaelyser is not used to estimate ferritin.

2. Again, in line 125 – 127, the authors write “Iron deficiency anemia is characterized by microcytic and hypochromic red blood cells. Out of the sample, 53.95% had microcytosis while hypochromia based on MCH was found in 37.5%.” This is a faulty assumption that only iron deficiency causes hypochromic-microcytic picture. Please revise to correct this.

3. Figures 1 and 2 appear the same; can the authors clarifies the differences? All the figures in panel 1 are the same as that in panel 2.

4. Table 1, 2 and 3, as well as figure 1 should be re-labelled taking into consideration that the sampling sites were different i.e. capillary (Haemocue) vs venous (automated analyser). This different sampling sites definitely leads to differences in the values recorded.

5. The axis labelling on the figure 3 suggest that Haemocue Hb – aHb = 0.36; however, the authors write in line 170 – 171 that “The bias between measurements is 0.36 ± 0.93 g/dL with the automated Hb yielding higher values”. If the data actually represents what the authors have stated, then the y-axis labelling should be revised.

6. In line 175 – 180, what precisely is the ROC analyses seeking to achieve?

DISCUSSION

1. I am unable to objectively assess the discussion since the faulty assumptions used for the data analyses has been carried into the data interpretation.

2. For example, in line 193 – 194, the authors write “classifying those with Hb values close to the threshold (12 g/dL). Women with ferritin values above 100 ug/L which generally excludes iron deficiency anemia [24] were regarded as anemics when using the POC device.” And based this on the Haemocue Hb estimation.

Reviewer #2: Comments

Study Title: “Accuracy of the portable Hemoglobin Analyzer HemoCue301 in Women of Reproductive Age in a Deprived Region of Northern Peru: An On-Field study”

Summary/observation

The study is very significant as it addresses an important issue in public health. The study aimed to assess the suitability and accuracy of a portable device called HemoCue301 which is used to classify anemia. While the study appreciates the use of HemoCue301, it advised that, the used of the device for the classification of iron deficiency anemia (IDA) should be done with caution.

Introduction

1. Authors should kindly take note and proofread the writing. Example, authors may take a second look at line 56 and 57 of page 3

Method

2. No justification was provided for the exclusion of heavy smokers from the study, line 82 page 5.

3. How were the blood samples transported to the laboratory where automated analyzer was used? Line 90 to 96 of page 5

4. Quality control issues were not indicated

6. PLOS authors have the option to publish the peer review history of their article (what does this mean?). If published, this will include your full peer review and any attached files.

Reviewer #1: **Yes: **Patrick Adu

Reviewer #2: **Yes: **David Larbi Simpong

---

## [Author Response · Author response to Decision Letter 0]

28 Sep 2023

Thank you for your time and suggestions.

Below we provide a detailed answer to the concerns raised. 

Answer to reviewers

Reviewer 1

General comment

Alarcón-Yaquetto et al have attempted to add data to the venous vrs capillary haemoglobin

comparison study viz-a-viz different devices/instrument conundrum. However, the authors have

not clearly defined basic concepts well to enable the reader to properly situate the findings. For

example, the title is a bit misleading considering the study design employed. In line 86 – 88

(page 5), the authors write that “Briefly, 301 cuvettes were filled with a drop of blood from the

middle finger fingertip. The first 2 drops of blood were discarded. The microcuvette was put in

the cuvette holder and a measurement was done immediately”. In line 90 – 91 (page 5), the

authors write that “Upon recruitment, a trained phlebotomist took venous samples of each

woman with the BD vacutainer system”. The laboratory samples were venous blood samples.

Therefore, the study compared venous haemoglobin (from automated analyser) to capillary

haemoglobin (Hemocue 301). it is a known fact that venous haemoglobin and capillary

haemoglobin are not equivalent (https://www.mdpi.com/2075-4418/12/12/3191;

10.1371/journal.pone.0278350). The title of the study is thus misleading and should be modified

to reflect exactly what was done. Again, the distinction between iron deficiency anaemia and

anaemia has not been made; the authors give the impression that the two terms are synonymous.

However, that is not a scientific fact. This has confounded the write up and the data analyses as

well as its interpretation. The manuscript should thus be revised holistically to address these.

Answer: Thank you for your comments. We have modified the title and we agree, one of the biggest problems in Peru is the generalisation of anaemia as IDA, we made changes throughout the document to reflect the distinction between anaemia and IDA. 

Abstract

1. The abstract has been written on the basis of the assumptions that were used to analyse

the data. As suggested above and below, the abstract need a re-work after taking the

suggested revisions into consideration.

Answer: The abstract has been modified accordingly.

2. In the objective statement, indicate that the sampling site was different for the Haemocue

and the automated haematology analyser.

Answer: We apologise for the confusion. The sampling site was the same. Once a participant fitted eligibility criteria and had a fingerstick haemoglobin reading between 8 and 12 g/dL, a venous sample was taken from her at the same site. We transported the samples to the laboratory where further tests were analysed. We clarified this on the manuscript. 

Introduction

1. Line 40 – 41, authors seem to portray IDA as being equivalent to anaemia by the

statement that “Peru is one of the countries where IDA is a severe public health problem

as shown by national statistics.”

Answer: We have rephrased the paragraph. This is how anaemia is treated in Peru. Even when only point of care devices such as the hemocue are used to diagnose the condition, iron supplementation is given. Confirmatory tests are hardly ever offered to patients living at remote areas. 

2. Line 56 – 57, authors write “Handheld devices have been routinely used to screen for

IDA in LMIC countries such as the HemoCue system” to imply that Hemocue is a device

for undertaking iron studies. This is misleading since the device estimates haemoglobin

levels which is different from ferritin, iron, TIBC etc.

Answer: We have changed IDA for anaemia. 

3. Authors again write that “The main objective of this study is to assess the suitability and

accuracy of this device as a method for IDA diagnosis in women of reproductive age in a

coastal city with resource constraints” Hemocue is not meant for iron studies.

Answer: We have changed IDA for anaemia and emphasise than even when hemocue is not meant for iron studies, it is used as such in Peru as it is the main device used in nation wide field studies to get national rates of anemia which in turn drive health policies.

Exclusion criteria

1. When authors write “Exclusion criterion was amenorrhea non explained by pregnancy

and heavy smoking” does this mean that both conditions should be present before

exclusion?

Answer: No. Both were different criteria to exclude. We have rephrased the paragraph. 

Statistical analyses

1. Authors defined Microcytosis using the CBC reported mean corpuscular values (MCV) <

80 fl.,. and hypochromia by mean corpuscular hemoglobin (MCH) values below 27. There are

many causes of microcytic-hypochromic red blood cell picture. With the emphasis the

authors are placing on iron deficiency anaemia, the authors should have at least

undertaken peripheral blood morphology evaluation to assess whether there were unique

red cell presentations such as pencil-shaped cells. This was not done.

Answer: The CBC analyses followed the ISLH´s guidelines. Briefly, slide review was performed in to assess morphological changes of the red blood cell. No thalassemia nor other type of hemoglobinopathies were found. Historically, prevalence of these type of genetic disorders is very low in Peru. We took measures to exclude subjects with underlying conditions and excluded subjects with inflammation according to proxies. We have updated information in the manuscript to reflect this information.

3. Authors should clearly differentiate iron deficiency (ferritin less than 15 ng/dl) from iron

deficiency anaemia (ferritin less than 15 ng/dl + Hb below threshold). This distinction has

not been made. Since the authors make the erroneous assumption that IDA is equivalent

to anaemia, it is not clear what the ROC analyses sought to predict.

Answer: Thank you for your suggestion. We have rephrased the methods section accordingly. 

4. Moreover, authors write that “Due to being an acute phase protein, ferritin is sensitive to

inflammation. Values above 120 ug/L were excluded from the analysis.” The CRP cut-off

of (5 mg/l) should be used to screen all the data but not only participants with ferritin

levels >120 mcg/L. It is even possible that some with ferritin levels >120 mcg/L had

haemochromatosis instead of inflammation-induced increased ferritin levels. Therefore,

the assumption of excluding data only based on the ferritin levels >120 mcg/L is

erroneous and should be corrected.

Answer: We have now included sensitivity analysis using CRP value of 5 mg/L as suggested. 

Results

1. The authors write that “123 women (40.2%) had ferritin values 125 above 15 g/dL using

the automated analyzer”. What does this represent? ID or anaemia? The two are not

equivalent and must be acknowledged as such. The haematotology anaelyser is not used

to estimate ferritin.

Answer: Thank you for noticing the mistake. 123 women had automated hemoglobin values above 12 g/dL. We meant to provide how this number compared to low ferritin values. We have updated the paragraph and now provide correct percentages.

2. Again, in line 125 – 127, the authors write “Iron deficiency anemia is characterized by

microcytic and hypochromic red blood cells. Out of the sample, 53.95% had microcytosis

while hypochromia based on MCH was found in 37.5%.” This is a faulty assumption that

only iron deficiency causes hypochromic-microcytic picture. Please revise to correct this.

Answer: While its true that iron deficiency is not the sole cause of hypochromic and microcytic red blood cells, the other causes are not prevalent in Peru. As we have updated in the methods section, thalassemia was checked as per ISLH guidelines. We have rephrased the paragraph and added in the discussion a sentence noting that iron deficiency is not the sole cause of hypochromic and microcytic RBCs.

3. Figures 1 and 2 appear the same; can the authors clarifies the differences? All the figures

in panel 1 are the same as that in panel 2.

Answer: The figures have been updated.

4. Table 1, 2 and 3, as well as figure 1 should be re-labelled taking into consideration that

the sampling sites were different i.e. capillary (Haemocue) vs venous (automated

analyser). This different sampling sites definitely leads to differences in the values

recorded.

Answer: We have updated the labelling as suggested. 

5. The axis labelling on the figure 3 suggest that Haemocue Hb – aHb = 0.36; however, the

authors write in line 170 – 171 that “The bias between measurements is 0.36 ± 0.93 g/dL

with the automated Hb yielding higher values”. If the data actually represents what the

authors have stated, then the y-axis labelling should be revised.

Answer: Thank you. We have corrected the y-axis label.

6. In line 175 – 180, what precisely is the ROC analyses seeking to achieve?

Answer We sought to analyse the accuracy of both methods in identifying a woman with ferritin values below 15 ng/dL. While the Hemocue is not meant for iron studies, that is the way it is used in Peru. This ROC analysis shows that the way the Peruvian government is tackling anemia by providing universal iron supplementation is an oversimplification of the condition. We run this analysis in the whole sample and in the sample excluding those with CRP values above 5 mg/l as suggested.

DISCUSSION

1. I am unable to objectively assess the discussion since the faulty assumptions used for the

data analyses has been carried into the data interpretation. For example, in line 193 – 194, the authors write “classifying those with Hb values close

to the threshold (12 g/dL). Women with ferritin values above 100 ug/L which generally

excludes iron deficiency anemia [24] were regarded as anemics when using the POC

device.” And based this on the Haemocue Hb estimation.

Answer: We have clarified the paragraph in the sense of noting Hemocue is a screening device and is not meant for iron deficiency diagnosis. However, that is the way it is being used in Peru. 

Reviewer 2

Summary/observation

The study is very significant as it addresses an important issue in public health. The study aimed to assess the suitability and accuracy of a portable device called HemoCue301 which is used to classify anemia. While the study appreciates the use of HemoCue301, it advised that, the used of the device for the classification of iron deficiency anemia (IDA) should be done with caution.

Answer: Thank you for your comments. 

Introduction

1. Authors should kindly take note and proofread the writing. Example, authors may take a second

look at line 56 and 57 of page 3

Answer: The mentioned paragraph has been edited for clarity. 

Method

2. No justification was provided for the exclusion of heavy smokers from the study, line 82 page 5.

Answer: Smoking is associated with decreased hemoglobin levels although the exact physiological mechanism is not fully understood. It might be due to the compound effect of inflammation, oxidative stress and the alteration of the antithrombotic system (Malenica et al 2017). The WHO has different cut-off values to define anemia in smokers. We have updated the information in the methods section.

Malenica M, Prnjavorac B, Bego T, Dujic T, Semiz S, Skrbo S, Gusic A, Hadzic A, Causevic A. Effect of Cigarette Smoking on Haematological Parameters in Healthy Population. Med Arch. 2017 Apr;71(2):132-136. doi: 10.5455/medarh.2017.71.132-136.

3. How were the blood samples transported to the laboratory where automated analyzer was 

used? Line 90 to 96 of page 5

Answer: We have detailed how samples were transported. 

4. Quality control issues were not indcated

Answer: We updated the information.

---

## [Decision Letter · Decision Letter 1]

19 Oct 2023

PONE-D-23-21513R1Accuracy of HemoCue301 Portable Hemoglobin Analyzer for Anemia Screening in Capillary Blood from Women of Reproductive age in a Deprived Region of Northern Peru: An On-Field StudyPLOS ONE

Dear Dr. Alarcón-Yaquetto,

Thank you for submitting your manuscript to PLOS ONE. After careful consideration, we feel that it has merit but does not fully meet PLOS ONE’s publication criteria as it currently stands. Therefore, we invite you to submit a revised version of the manuscript that addresses the points raised during the review process.

Reviewer has made some very good additional points and I kindly ask the authors to address all of them. Please submit your revised manuscript by Dec 03 2023 11:59PM. If you will need more time than this to complete your revisions, please reply to this message or contact the journal office at plosone@plos.org. Please include the following items when submitting your revised manuscript:A rebuttal letter that responds to each point raised by the academic editor and reviewer(s). You should upload this letter as a separate file labeled 'Response to Reviewers'.A marked-up copy of your manuscript that highlights changes made to the original version. You should upload this as a separate file labeled 'Revised Manuscript with Track Changes'.An unmarked version of your revised paper without tracked changes. You should upload this as a separate file labeled 'Manuscript'.If applicable, we recommend that you deposit your laboratory protocols in protocols.io to enhance the reproducibility of your results. Protocols.io assigns your protocol its own identifier (DOI) so that it can be cited independently in the future. For instructions see: https://journals.plos.org/plosone/s/submission-guidelines#loc-laboratory-protocols. Additionally, PLOS ONE offers an option for publishing peer-reviewed Lab Protocol articles, which describe protocols hosted on protocols.io. Read more information on sharing protocols at https://plos.org/protocols?utm_medium=editorial-email&utm_source=authorletters&utm_campaign=protocols.

We look forward to receiving your revised manuscript.

Kind regards,

Benedikt Ley, PhD

Academic Editor

PLOS ONE

Journal Requirements:

Reviewers' comments:

Reviewer's Responses to Questions

**Comments to the Author**

1. If the authors have adequately addressed your comments raised in a previous round of review and you feel that this manuscript is now acceptable for publication, you may indicate that here to bypass the “Comments to the Author” section, enter your conflict of interest statement in the “Confidential to Editor” section, and submit your "Accept" recommendation.

Reviewer #1: (No Response)

Reviewer #2: All comments have been addressed

2. Is the manuscript technically sound, and do the data support the conclusions?

Reviewer #1: Partly

Reviewer #2: Yes

3. Has the statistical analysis been performed appropriately and rigorously? 

Reviewer #1: Yes

Reviewer #2: Yes

4. Have the authors made all data underlying the findings in their manuscript fully available?

Reviewer #1: Yes

Reviewer #2: Yes

5. Is the manuscript presented in an intelligible fashion and written in standard English?

Reviewer #1: No

Reviewer #2: Yes

6. Review Comments to the Author

Reviewer #1: Summary

The authors have made attempt to address the issues that were raised in the initial review process. While the effort is commended, there are still some issues that the authors need to address to make the manuscript technically sound. Authors still make it look like IDA is equivalent to anaemia. While this might be an accepted misconception in Peru (as the authors claim), the manuscript is written for the global audience and should therefore be approached as such. Additionally, the study design was such that fingertip blood (capillary blood) was used for the Hemocue Hb estimation; venous blood was used for the automated analyser Hb estimation. This should be consistently maintained throughout the manuscript to provide context for the data interpretation since the two blood samples are not equivalent.

ABSTRACT

Methods: in the sentence (line 22) “Adult women with HemoCue 301 readings below 12 g/dL were recruited”, authors should indicate that this was capillary blood.

In line 26 (Results), authors should specify that capillary blood was used for Hemocue Hb in the sentence “The Hemocue 301 has a bias of 0.36 ± 0.93 g/dL respect to the automated Hb.”

INTRODUCTION

Paragraph 2 of the introduction, authors correctly use the public health significance of anaemia in their classification (41 – 43). However, in paragraph 3 (line 44 – 45), the authors introduce a dimension that the severe anaemia challenge (prevalence of 40% in Peru) is questionable on the supposition that haemoglobin measurement is only a proxy for IDA. IDA classification is based on iron parameter estimates but not solely on haemoglobin measurement. Authors should decouple these in the write up.

The reference 9 cited in the statement “Furthermore, automated hematology analyzers —the method suggested by the WHO [9]” line 54 is not a WHO policy document; please correct this.

METHODS

The study design was such that fingertip blood (capillary blood) was used for the Hemocue Hb estimation; venous blood was used for the automated analyser Hb estimation. This should be consistently maintained throughout the manuscript to provide context for the data interpretation since the two blood samples are not equivalent. Therefore,

under the inclusion criteria, authors should specify in line 80 – 81 that the Hemocue Hb was estimated from capillary blood.

Under laboratory analyses (line 95), authors should specify that CBC was undertaken using venous blood.

In line 96, the reference 18 as stated by the authors in the main reference list is not an ISLH guideline; please rectify this.

Under laboratory analyses (line 96 – 97), authors seem to make an erroneous claim that peripheral blood evaluation alone can be used rule out thalassaemia or other types of haemoglobinopathies. This is haematologically inaccurate. Thalassaemia trait could be silent and may present with few target cells only in peripheral smear. These same target cells are present in iron deficiency, haemoglobin C disease etc. authors should correct this as a peripheral blood morphology evaluation cannot be used to definitively rule out thalassaemia or other haemoglobinopathies.

Statistical analyses

In line 118 – 119, authors are not clear as to what the ROC sought to determine. The accuracy in detecting what exactly? is this ferritin <15? From the manuscript as a whole, one get the impression that the ROC was undertaken to provide a cut-off for Hemocue Hb or analyser Hb that could predict higher likelihood of IDA. If this is what was the target, the authors need to set out all the assumptions made under the statistical analyses.

RESULTS

In line 141, in writing “We further classified participants based on tertiles of both HemoCue 301 and automated Hb”, authors should specify that capillary blood was used for Hemocue Hb vs venous blood for analyser Hb.

In table 2, authors should include in the caption capillary blood (Hemocue Hb) and venous blood (analyser Hb).

In line 183, specify the venous vs capillary blood in the sentence

In line 189 – 196, the ROC & AUC should have a cut-off to guide clinical utility. In addition, there should be other parameters like PPV, NPV, sensitivity and specificity. These are better captured in a separate Table to inform the reader of the reliability of the estimates.

DISCUSSION

The discussion is silent on perhaps the key highlight of the study; the ROC and AUC data which sought to establish a cut-off for Hemocue Hb and analyser Hb that could be used as a surrogate means of diagnosing iDA in a resource poor setting. The rational for undertaking AUC and cut-off for analytes particularly in poor settings is to provide clinicians with surrogate estimates that provide high degree of suspicion so that treatment could be prescribed even in the absence of specific biochemical tests for diagnosing IDA. However, this important diagnostic dimension of the data is not explored in the discussion which leaves an obvious question as to why the authors employed the ROC in the first place.

Additionally, authors should not be silent on the fact that there are two dimensions on the results; 1) differences in methods [Hemocue vs automated analyser] and 2) different sampling sites of blood [capillary blood for Hemocue Hb vs venous blood for analyser Hb]. These two dimensions should be kept in focus as each affect the results. Should another lab repeat this study by maintain the two measurement techniques (Hemocue vs analyser) but use the same source of blood, the results will not be directly comparable to this study. Thus, it is important that authors stress these in the discussion.

MINOR

Line 95 “blood count (CBC) was performed withing two hours of the sample obtention” please correct “withing” and “obstention” and these do not exist.

Line 238 the deploy in “Another argument used for the widespread deploy of HemoCue” to deployment.

Reviewer #2: Authors have adequately addressed all comments raised in the previous round of review and that this manuscript is now acceptable for publication

7. PLOS authors have the option to publish the peer review history of their article (what does this mean?). If published, this will include your full peer review and any attached files.

Reviewer #1: **Yes: **Dr Patrick Adu

Reviewer #2: **Yes: **David Larbi Simpong

---

## [Author Response · Author response to Decision Letter 1]

21 Oct 2023

Replies

Summary

The authors have made attempt to address the issues that were raised in the initial review process. While the effort is commended, there are still some issues that the authors need to address to make the manuscript technically sound. Authors still make it look like IDA is equivalent to anaemia. While this might be an accepted misconception in Peru (as the authors claim), the manuscript is written for the global audience and should therefore be approached as such. Additionally, the study design was such that fingertip blood (capillary blood) was used for the Hemocue Hb estimation; venous blood was used for the automated analyser Hb estimation. This should be consistently maintained throughout the manuscript to provide context for the data interpretation since the two blood samples are not equivalent. 

ABSTRACT

Methods: in the sentence (line 22) “Adult women with HemoCue 301 readings below 12 g/dL were recruited”, authors should indicate that this was capillary blood. 

Reply: Thank you. We have modified the sentence as suggested.

In line 26 (Results), authors should specify that capillary blood was used for Hemocue Hb in the sentence “The Hemocue 301 has a bias of 0.36 ± 0.93 g/dL respect to the automated Hb.”

Reply: We have modified the sentence as suggested.

INTRODUCTION

Paragraph 2 of the introduction, authors correctly use the public health significance of anaemia in their classification (41 – 43). However, in paragraph 3 (line 44 – 45), the authors introduce a dimension that the severe anaemia challenge (prevalence of 40% in Peru) is questionable on the supposition that haemoglobin measurement is only a proxy for IDA. IDA classification is based on iron parameter estimates but not solely on haemoglobin measurement. Authors should decouple these in the write up.

Reply: We have added a line stating that IDA diagnosis needs the estimation of iron parameters.

The reference 9 cited in the statement “Furthermore, automated hematology analyzers —the method suggested by the WHO [9]” line 54 is not a WHO policy document; please correct this.

Reply: Thank you for noticing the error. The correct reference is given in the updated version.

METHODS

The study design was such that fingertip blood (capillary blood) was used for the Hemocue Hb estimation; venous blood was used for the automated analyser Hb estimation. This should be consistently maintained throughout the manuscript to provide context for the data interpretation since the two blood samples are not equivalent. Therefore, under the inclusion criteria, authors should specify in line 80 – 81 that the Hemocue Hb was estimated from capillary blood. Under laboratory analyses (line 95), authors should specify that CBC was undertaken using venous blood. In line 96, the reference 18 as stated by the authors in the main reference list is not an ISLH guideline; please rectify this.

Reply: Line 80-81, 95 and the ISLH reference have been modified/added. 

Under laboratory analyses (line 96 – 97), authors seem to make an erroneous claim that peripheral blood evaluation alone can be used rule out thalassaemia or other types of haemoglobinopathies. This is haematologically inaccurate. Thalassaemia trait could be silent and may present with few target cells only in peripheral smear. These same target cells are present in iron deficiency, haemoglobin C disease etc. authors should correct this as a peripheral blood morphology evaluation cannot be used to definitively rule out thalassaemia or other haemoglobinopathies.

Reply: The paragraph has been rephrased and we added in the limitation section the fact that blood morphology evaluation cannot be used to rule out hemoglobinopathies. 

Statistical analyses

In line 118 – 119, authors are not clear as to what the ROC sought to determine. The accuracy in detecting what exactly? is this ferritin <15? From the manuscript as a whole, one get the impression that the ROC was undertaken to provide a cut-off for Hemocue Hb or analyser Hb that could predict higher likelihood of IDA. If this is what was the target, the authors need to set out all the assumptions made under the statistical analyses.

Reply: The accuracy to detect values <15. We have updated this in the article. We updated methods about estimation of optimal cut-off value. 

RESULTS

In line 141, in writing “We further classified participants based on tertiles of both HemoCue 301 and automated Hb”, authors should specify that capillary blood was used for Hemocue Hb vs venous blood for analyser Hb.

Reply: We have updated the line as suggested. 

In table 2, authors should include in the caption capillary blood (Hemocue Hb) and venous blood (analyser Hb).

Reply: We have updated the table as suggested. 

In line 183, specify the venous vs capillary blood in the sentence.

Reply: We have updated the line as suggested. 

In line 189 – 196, the ROC & AUC should have a cut-off to guide clinical utility. In addition, there should be other parameters like PPV, NPV, sensitivity and specificity. These are better captured in a separate Table to inform the reader of the reliability of the estimates.

Reply: We have added table 4 which provides PPV, NPV, sensitivity and specificity. We updated the methods section respectively.

DISCUSSION

The discussion is silent on perhaps the key highlight of the study; the ROC and AUC data which sought to establish a cut-off for Hemocue Hb and analyser Hb that could be used as a surrogate means of diagnosing IDA in a resource poor setting. The rational for undertaking AUC and cut-off for analytes particularly in poor settings is to provide clinicians with surrogate estimates that provide high degree of suspicion so that treatment could be prescribed even in the absence of specific biochemical tests for diagnosing IDA. However, this important diagnostic dimension of the data is not explored in the discussion which leaves an obvious question as to why the authors employed the ROC in the first place.

Reply: We have added table 4 results in the discussion.

Additionally, authors should not be silent on the fact that there are two dimensions on the results; 1) differences in methods [Hemocue vs automated analyser] and 2) different sampling sites of blood [capillary blood for Hemocue Hb vs venous blood for analyser Hb]. These two dimensions should be kept in focus as each affect the results. Should another lab repeat this study by maintain the two measurement techniques (Hemocue vs analyser) but use the same source of blood, the results will not be directly comparable to this study. Thus, it is important that authors stress these in the discussion. 

Reply: We have updated the discussion accordingly.

MINOR

Line 95 “blood count (CBC) was performed withing two hours of the sample obtention” please correct “withing” and “obstention” and these do not exist.

Line 238 the deploy in “Another argument used for the widespread deploy of HemoCue” to deployment.

Reply: We have amended the referenced lines.

---

## [Editor Report · Decision Letter 2]

24 Oct 2023

Accuracy of HemoCue301 Portable Hemoglobin Analyzer for Anemia Screening in Capillary Blood from Women of Reproductive age in a Deprived Region of Northern Peru: An On-Field Study

PONE-D-23-21513R2

Dear Dr. Alarcón-Yaquetto,

We’re pleased to inform you that your manuscript has been judged scientifically suitable for publication and will be formally accepted for publication once it meets all outstanding technical requirements.

Kind regards,

Benedikt Ley, PhD

Academic Editor

PLOS ONE
---

## [Editor Report · Acceptance letter]

5 Nov 2023

PONE-D-23-21513R2 

Accuracy of HemoCue301 Portable Hemoglobin Analyzer for Anemia Screening in Capillary Blood from Women of Reproductive age in a Deprived Region of Northern Peru: An On-Field Study 

Dear Dr. Alarcón-Yaquetto:

I'm pleased to inform you that your manuscript has been deemed suitable for publication in PLOS ONE. Congratulations! Your manuscript is now with our production department. 

Kind regards, 

on behalf of

Dr Benedikt Ley 

Academic Editor

PLOS ONE